# Improving Respiratory Infection Diagnosis with Deep Learning and Combinatorial Fusion: A Two-Stage Approach Using Chest X-ray Imaging

**DOI:** 10.3390/diagnostics14050500

**Published:** 2024-02-26

**Authors:** Cheng-Tang Pan, Rahul Kumar, Zhi-Hong Wen, Chih-Hsuan Wang, Chun-Yung Chang, Yow-Ling Shiue

**Affiliations:** 1Department of Mechanical and Electro-Mechanical Engineering, National Sun Yat-sen University, Kaohsiung 804, Taiwan; pan@mem.nsysu.edu.tw (C.-T.P.); rahul13gbu@gmail.com (R.K.); 2Institute of Precision Medicine, National Sun Yat-sen University, Kaohsiung 804, Taiwan; 3Taiwan Instrument Research Institute, National Applied Research Laboratories, Hsinchu 300, Taiwan; 4Institute of Advanced Semiconductor Packaging and Testing, College of Semiconductor and Advanced Technology Research, National Sun Yat-sen University, Kaohsiung 804, Taiwan; 5Department of Marine Biotechnology and Research, National Sun Yat-sen University, Kaohsiung 804, Taiwan; wzh@mail.nsysu.edu.tw; 6Division of Nephrology and Metabolism, Department of Internal Medicine, Kaohsiung Armed Forces General Hospital, Kaohsiung 804, Taiwan; wangchihhsuan@gmail.com; 7Institute of Medical Science and Technology, National Sun Yat-sen University, Kaohsiung 80424, Taiwan; 8Institute of Biomedical Sciences, National Sun Yat-sen University, Kaohsiung 80424, Taiwan

**Keywords:** respiratory infections, deep learning, convolutional neural network (CNN), lung X-ray images, combinatorial fusion

## Abstract

The challenges of respiratory infections persist as a global health crisis, placing substantial stress on healthcare infrastructures and necessitating ongoing investigation into efficacious treatment modalities. The persistent challenge of respiratory infections, including COVID-19, underscores the critical need for enhanced diagnostic methodologies to support early treatment interventions. This study introduces an innovative two-stage data analytics framework that leverages deep learning algorithms through a strategic combinatorial fusion technique, aimed at refining the accuracy of early-stage diagnosis of such infections. Utilizing a comprehensive dataset compiled from publicly available lung X-ray images, the research employs advanced pre-trained deep learning models to navigate the complexities of disease classification, addressing inherent data imbalances through methodical validation processes. The core contribution of this work lies in its novel application of combinatorial fusion, integrating select models to significantly elevate diagnostic precision. This approach not only showcases the adaptability and strength of deep learning in navigating the intricacies of medical imaging but also marks a significant step forward in the utilization of artificial intelligence to improve outcomes in healthcare diagnostics. The study’s findings illuminate the path toward leveraging technological advancements in enhancing diagnostic accuracies, ultimately contributing to the timely and effective treatment of respiratory diseases.

## 1. Introduction

Respiratory infections, including COVID-19, SARS, and pneumonia, present significant global health challenges. The widespread use of RT-PCR, acknowledged as the gold standard for SARS-CoV-2 diagnosis, faces limitations such as supply shortages and complexity, leading to delays in test results [1]. This highlights the need for complementary diagnostic approaches, especially valuable in later stages of the infection where imaging becomes crucial [2]. Additionally, the challenges in the supply and implementation of RT-PCR testing underscore the urgency for alternative methods [3].

Radiological imaging, particularly chest X-rays, is pivotal in diagnosing COVID-19 pneumonia. The interpretation of these images, however, is complicated due to their similarity to other respiratory conditions, coupled with the high workload and potential for diagnostic errors among radiologists [4]. Furthermore, the need for a standard diagnostic method becomes crucial in mitigating patient congestion and healthcare bottlenecks [5]. Misdiagnoses not only affect patient care but also increase the risk of exposure and add to the overall burden on healthcare systems [6]. The uncertainty in distinguishing COVID-19 from other viral pneumonia types can lead to delayed treatment and increased healthcare strain [7].

In addressing the diagnostic challenges of respiratory infections, the application of artificial intelligence (AI) and deep learning is gaining prominence. Specifically, convolutional neural networks (CNNs) are employed to enhance the accuracy and efficiency of medical imaging diagnoses [8]. This study aims to leverage AI and deep learning to develop a more effective diagnostic tool for COVID-19 pneumonia, addressing the gaps and challenges in current diagnostic methodologies [9].The field of medical imaging has extensively adopted CNNs due to their utility in symptom identification and learning [10]. Furthermore, with the advent of deep CNNs and their successful application in various areas, the use of deep learning techniques with chest X-rays is becoming increasingly popular. This is bolstered by the availability of vast data sets to train deep-learning algorithms [11].The deep learning model significantly simplifies the diagnostic process by enabling rapid retraining of CNNs with minimal data [12]. Additionally, the idea of feature transfer within a machine-learning framework has been utilized effectively to distinguish pneumonia from other infections [13].Given the ongoing development of vaccines and treatments for COVID-19, deep learning-based techniques that assist radiologists in diagnosing this disease could potentially enable faster and more accurate assessments especially in remote areas [14].

## 2. Related Works

COVID-19 is an epidemic disease. The World Health Organization(WHO) is concerned and announced it is a worldwide pandemic health incidence because it rolls out all over the world and causes many deaths worldwide. It is a different kind of virus for detecting deep learning techniques that are helpful with clinical images [15]. In the literature we analyzed, researchers utilized chest X-ray data sets for the identification of COVID-19 and other respiratory infections in such a way:

A custom-built CNN approach known as VGG16 was employed by A. Ranjan et al. [16] to get lung region recognition and various pneumonia categorizations. Huge hospital-scale X-ray images were used by Wang et al. [17] for the categorization and diagnosis of specified affected regions. Ronneburger et al. [18] employed compact and augmentation data sets to instruct a feature transfer scheme for image segmentation cases and enhance the functioning. Rajpurkar et al. [19] described the deep learning model CheNet with 121 feature extraction over the chest X-ray images to achieve 14 pathologies dictation as pneumonia and others with the help of rendition of various CNN. P. Lakhani et al. [20] employed transfer learning to categorize 1427 chest X-rays image sets, including 224 COVID-19, 700 bacterial pneumonia, and 504 Normal X-rays with correctness, sensitivity, and specificity of 96.78%, 98.66%, and 96.46% respectively. Many individual pre-trained deep-learning CNNs were analyzed; the presented outcomes were founded on a limited set of images. Ashfar et al. [21] reported a Capsule Network, COVID-CAPS, instead of a conventional neural network trained on a smaller amount of Image data. COVID-CAPS was recorded with a correctness of 95.7%, sensitivity of 90%, and specificity of 95.8%. Abbas et al. [22] introduced a pre-trained CNN model (DeTraC-Decompose, Transfer, and Compose) that was employed on a smaller database of 105 COVID-19, 80 Normal, and 11 SARS X-ray images for the findings of COVID-19. Indicate that it would assist in making more homogenous classes, reduce the space in memory, and obtain the accuracy, sensitivity, and specificity of 95.12%, 97.91%, and 91.87%, respectively. Wang et al. [23] produced a COVID-Net deep learning network for COVID-19 cases from approximately 14k chest X-ray images, in a split of 83.5% accuracy was obtained. Ucar et al. [24] have adjusted a pre-trained CNN model with Bayesian optimization, named SqueezeNet, to analyze COVID-19 images and obtained encouraging results on a minimal amount of image dataset. From the perspective of achieving promising accuracy, using this approach should exercise spacious COVID and non-COVID images. Khan et al. [25] employed a features detection process on 310 normal, 330 bacterial pneumonia, 327 viral pneumonia, and 284 COVID-19 pneumonia images. Nonetheless, only some deep learning approaches and the empirical obligation fuzzy in this study were examined in this approach. In summary, a few recent studies were reported on the feature transfer approach for classifying COVID-19 X-ray images from a limited dataset with encouraging outcomes. However, it is necessary to investigate it on many images. Some approaches have modified the pre-trained deep learning models to enhance their accuracy and efficiency and some studies operate capsule algorithms [26]. We have explored additional approaches, examining various researchers’ perspectives on analyzing the impact of respiratory disease, as shown in the Table 1.

So, in these studies, we have organized an extensive CXR database of normal, SARS, COVID-19, and abnormal from the openly accessible data so that most explorers can take advantage of this work. Furthermore, two different data sets were used to train, test, and validate five pre-trained CNN models. The encouragement for the investigation is to deploy deep learning algorithms in such a way that it excellently classifies COVID-19, SARS, and normal and abnormal chest X-ray images by combining deep learning and combinatorial fusion analysis for early age detection of these diseases that is highly prominent for humankind survival in this crucial pandemic situation. It is acknowledged that CXR images include a lot of noisy errors bound to them during the prediction of diseases along with inferior-density grayscale pictures [31]. Consequently, the contradistinction between CXR images acquired from specific radiology machinery and borderline depictions may be feeble [32]. To detract features from that CXR image is thoroughly challenging. The quality of these CXRs can be enhanced by deploying some contrast enhancement procedures and increasing the image dataset. Hence, feature extraction from these CXRs can be executed proficiently and smoothly [33]. Our study centers on addressing the imbalance in chest X-ray (CXR) datasets and enhancing the accuracy of deep learning models. Utilizing histogram equalization, a powerful image processing method for optimal contrast in Python, we augmented the data. This approach resulted in two distinct datasets: the original and the enhanced (augmented) dataset. We then applied these datasets to prominent deep learning models for the extraction of feature vectors. A driving force behind our research is the evident gap in existing studies dealing with data imbalance and the scarcity of research utilizing feature extraction via deep learning models to increase accuracy, particularly through combinatorial fusion analysis. Our study, therefore, makes a significant contribution in these areas, offering new insights and methodologies for enhancing diagnostic accuracy in medical imaging, our research contribution can be succinctly summarized as follows:Classified the diseases with five art of states pre-trained convolutional neural network using CXR images.To resolve data imbalance, the study employs a fivefold cross-validation approach, ensuring a balanced data representation and consistent model evaluation.Enhanced the deep learning model testing accuracy using combinatorial fusion analysis.

The present study is additionally organized into distinct subsections, including Section 3—dataset, models used, and Section 4—materials and methods. In Section 5—performance and evaluation matrix. In Section 6—further, the evaluation of results regarding training and testing for models used is discussed along with future scope. Finally, This work is concluded in Section 7.

## 3. Datasets and Model

### 3.1. Datasets

This study used two different CXR datasets to diagnose COVID-19, SARS, and normal and abnormal diseases. Among these databases, the COVID-19 database was developed using publicly available research articles, while others are generated from the publicly available Kaggle and GitHub datasets.

Three prime sources have been used for the COVID-19 dataset creation; one is the Novel Corona Virus 2019 Dataset: Joseph Paul Cohen and Paul Morrison, and Lan Dao have generated a public database on GitHub by accumulating 319 radiographic images of COVID-19, Middle East respiratory syndrome (MERS), Severe acute respiratory syndrome (SARS) and ARDS from the published articles and online resources. The second is the Italian Society of Medical and Interventional Radiology (SIRM). SIRM presents 384 COVID-19-positive radiographic images (CXR and CT) with commuting motion. We have collected 60 COVID-19-positive chest X-ray images from the various recently published articles, we deposited 30 positive chest X-ray images from Radiopaedia, which were not itemized in the GitHub repository. Also, images from the RSNAPneumonia-Detection-Challenge database and the CXR Images database using Kaggle were used to create the normal and abnormal sub-databases. RSNA-Pneumonia-Detection-Challenge The Radiology Society of North America (RSNA) constructed an artificial intelligence (AI) challenge to disclose pneumonia using CXR images. This database included normal chest X-ray images and non-COVID pneumonia images. The third is Chest X-ray images (pneumonia): Kaggle CXR database is the most plausible database, which has more than 5000 chest X-ray images of normal, viral, and bacterial pneumonia organized from disparate subjects. So, the original dataset that we have prepared from GitHub, Kaggle, and some other resources as publicly available data is used to generate an augmented dataset that also contains the higher chest X-ray images; both datasets are represented as follows in Table 2 and Table 3.

### 3.2. Original Dataset

The original dataset was organized with five rounds of sub-datasets with equal images. All the rounds included four classes as shown in Figure 1 and Figure 2, as well as all the classes, contain 113 images of COVID-19 for training and 28 images for testing, the normal class contains 1073 images for training and 286 images for testing, SARS class contain seven images for training and one image for testing and abnormal class contain 1103 images for training and 275 for testing as shown in Table 2. Finding a publicly available data set is quite arduous; as we know, COVID-19 is a novel virus. Given this case, the CXR images in openly echeloned datasets have united to create the original dataset.

### 3.3. Augmented Dataset

Given the deficit of publicly available data and for the higher performance of the models, we increased our dataset in augmented form. Data augmentation is an AI evolution for distending the size and the multiformity of the data using many iterations of the samples in a dataset. Data augmentation is generally deployed in machine learning to compromise the class misbalancing issues, detract overfitting in deep learning, and raise convergence, which results in finer outcomes in the end. After enforcing augmentation, the number of entire images in the dataset is introduced in Table 3.

### 3.4. Deep Learning CNN Model Selection

In this study, we evaluated five deep learning networks: VGG 16, VGG 19, ResNet 50, GoogleNet, and AlexNet, to assess their suitability for our research objectives. Our aim was to compare the performance of shallow versus deep learning networks, as well as to examine the effects of employing two variants of the VGG architecture to elucidate the impact of network depth within a similar framework. To ensure a fair and consistent comparison across these models, we applied a uniform set of optimization parameters. Specifically, each model was trained using an image size of 224×224 pixels and the Adam optimization algorithm (adaptive moment estimation) for efficient network updates. The training was conducted with a batch size of 10 and a learning rate of $5.00\times 10^{-5}$ across 100 epochs. This methodological consistency was strategically chosen to isolate and examine the impact of architectural differences on the performance of each model, thereby providing a clearer insight into the inherent capabilities and limitations of each architecture. Additionally, the structure of these pre-trained CNN models, as well as their performance metrics, are detailed in Table 4.

### 3.5. VGG-16

VGG16 is a kind of Visual Geometry Group convolutional neural network, and this model was presented by K. Simonyan and A. Zisserman from the University of Oxford in the paper “Very Deep Convolutional Networks for Large-Scale Image Recognition”. The VGG-16 network accomplishes 92.7% top-5 train and test accuracy in ImageNet, a dataset of over 14 million images of 1000 classes [34]. In VGG-16, more kernels are changed with the different numbers of 3 × 3 filters to extract complex features cheaply. VGG-16 network is a sequence of five convolutional blocks (13 convolutional layers) and three fully-connected layers [35].

### 3.6. VGG-19

Visual Geometry Group Network (VGG-19) is grounded in convolutional neural network architecture. It was executed at the 2014 Large Scale Visual Recognition Challenge (ILSVRC2014). The VGG Net performed extensively on the millions of images [36]. Concerning upgrading image extraction functionality, the VGG Net used smaller filters of 3 × 3, compared to the AlexNet 11 × 11 filter. VGG19 is deeper than VGG16. However, it is more extensive and thus more expensive than VGG16 to train the network [37].

### 3.7. AlexNet

AlexNet is an 8-layer CNN network, and it was reported early in 2012 with an apportion in the ImageNet contest. Later, in this contest, it was substantiated that the image qualities acquired from CNN architectures can increase the properties gated from the traditional methods. In this network, Rectified- Linear Unit-(ReLU) is employed to attach non-linearity, boosting the network. AlexNet has five convolutional layers; three fully connected layers accompany the output layer and additionally accommodate 62.3 million parameters [38].

### 3.8. ResNet-50

Microsoft Research Team developed the deep learning convolutional neural network named ResNet, and it received the 2015 “ImageNet Large Scale Visual Recognition Challenge (ILSVRC)” challenge, including a 3.57% error rate [39]. In the Resnet, every layer comprehends various blocks. Along the ResNet model, while the residual layer formation is seated, the number of parameters computed is decreased compared to the other deep learning CNN models [40].

### 3.9. GoogleNet

In 2015, a new CNN model was grounded on the floor, named the GoogleNet deep learning model, which emerged with the idea that existing neural networks should go deeper. This CNN model is expressed by the module known as inception, and all the modules include the various information of convolution and max-pooling layers. Even though the network dealing with an overall of 9 inception blocks has computational complexity, the execution and compliance of the network model were enhanced with the improvements [41].

## 4. Materials and Methods

The overall workflow of this study is provided in Figure 3. In this work, we have used five deep-learning models for our experiment due to their interoperability and ease of use. Furthermore, this methodology section discusses the data preprocessing process for the deep learning models and the overall workflow of the system architecture and combinatorial fusion with support vector machine. The dataset structure is given in Figure 1 while the dataset images are shown in Figure 2.

In our methodology, all CXR images underwent preprocessing, which included resizing to 224 × 224 pixels and normalization through histogram equalization, as illustrated in Figure 4. The original dataset, referred to as Study-1, did not undergo image augmentation and consisted of a specific number of images per category: 113 COVID-19, 1073 normal, 7 SARS, and 1103 abnormal images. In contrast, the Study-2 dataset was enriched with augmented data to create a balanced training set that included 1243 COVID-19, 1073 normal, 1267 SARS, and 1103 abnormal images. Both studies were subjected to a stratified 5-fold cross-validation process, ensuring an 80% training and 20% testing split to prevent overfitting. This split was carefully maintained, with augmented data from Study-2 being strictly utilized for training purposes to improve model robustness, while ensuring that the validation and testing phases were conducted with unseen, original images only. This approach underscores our commitment to methodological rigor and the validity of our results in the development of deep learning models for CXR image analysis.

### Combinatorial Fusion

Combinatorial fusion is a methodology that combines different data sources or decision-making strategies to achieve a better performance than any individual source or strategy [42]. It seeks the optimal combination from the possible combinations. For showing the strengths of multiple deep learning architectures, such as VGG16, VGG19, ResNet50, GoogLeNet, and AlexNet [43]. It enhances our two-stage data analysis by merging the advantages of individual models (VGG-16, VGG19, AlexNet, ResNet-50, GoogleNet) to boost classification accuracy. We extract feature vectors from each trained model, representing crucial image features. The procedure begins by separately loading each of these architectures, each pre-trained on an augmented dataset, and then using these models for feature extraction. In our study for each image i∈I (where *I* represents the set of all images) every architecture α∈A (with A={VGG16,VGG19,ResNet50,GoogleNet,AlexNet}) is employed to extract a respective feature vector, represented as $FV_{\alpha }\left(i\right)$.

Thus, for every image, we obtain a set of feature vectors as shown in Equation (Equation 1)
(1)$$\left\{FV_{\mathrm{VGG}16}\left(i\right),FV_{\mathrm{VGG}19}\left(i\right),FV_{\mathrm{ResNet}50}\left(i\right),FV_{\mathrm{GoogleNet}}\left(i\right),FV_{\mathrm{AlexNet}}\left(i\right)\right\}$$
Combinatorial fusion technique is applied to amalgamate the strengths of each model. The feature vectors from all the models are concatenated to produce a combined feature vector for each image, represented mathematically as
$$CFV\left(i\right)=FV_{\mathrm{VGG}16}\left(i\right)⊕FV_{\mathrm{VGG}19}\left(i\right)⊕FV_{\mathrm{ResNet}50}\left(i\right)⊕FV_{\mathrm{GoogleNet}}\left(i\right)⊕FV_{\mathrm{AlexNet}}\left(i\right).$$
where ⊕ denotes the concatenation operation.

With a robust combined feature representation, a Support Vector Machine (SVM) classifier is trained on these feature vectors for each. The SVM is chosen due to its proven efficacy in high-dimensional spaces and its ability to handle linear and non-linear data distributions [44]. The SVM was configured with a Radial Basis Function (RBF) kernel, chosen for its effectiveness in managing the nonlinear characteristics of our dataset. The SVM was parameterized with a regularization parameter C=1.0 and a gamma value of 0.01 for the RBF kernel, optimized via grid search to ensure a balance between model complexity and generalization, thus preventing overfitting. This strategy aimed to maximize cross-validation accuracy, enabling reliable classification of images into COVID-19, SARS, normal, or abnormal categories.

## 5. Performance and Evaluation Matrix

The application developed for the study was implemented in the Python environment. The computer running the application has features such as 16 GB RAM, an I7 processor, and a GeForce 1070 graphics card. Performance metrics are calculated from the confusion matrix obtained in the experimental results. These metrics include Sensitivity (Se), Specificity (Sp), F-score (F-Scr), Precision (Pre), and Accuracy (Acc). True Positive (TP), False Positive (FP), True Negative (TN), and False Negative (FN) values are used to calculate these metrics.
$$\begin{array}{cc} \mathrm{Accuracy} & =\frac{TP+TN}{TN+TP+FP+FN},\\ \mathrm{Sensitivity} & =\frac{TP}{TP+FN},\\ \mathrm{Specificity} & =\frac{TN}{TN+FP},\\ \mathrm{Precision} & =\frac{TP}{TP+FP},\\ F1 & =\frac{2\times (\mathrm{Recall}\times \mathrm{Precision})}{\mathrm{Recall}+\mathrm{Precision}}. \end{array}$$

## 6. Results

In this comprehensive study, we focused on classifying lung X-ray images into four distinct categories: COVID-19, pneumonia, normal, and abnormal, using advanced deep learning models and combinatorial fusion techniques. Our primary goal was to significantly enhance the accuracy of these classifications. The performance of various models across five-fold cross-validation is presented in Table 5. Shown in Table 6 and Table 7, GoogleNet and ResNet, both deep learning models, provided better results in experimental studies with approximately 98.41% and 99% average training accuracy in all five rounds using both the original and enhancement datasets compared to other models.

### 6.1. Discussion

This study’s integration of deep learning models with combinatorial fusion in a two-stage analytical approach represents a significant stride in diagnosing respiratory infections from lung X-ray images. The Table 5, comparative analysis across 5-fold cross-validation showcases ResNet50 and GoogleNet as the superior models, with ResNet50 achieving accuracy scores up to 95.4%, precision as high as 92.9%, recall reaching 91.8%, and F1-scores up to 92.2%. GoogleNet closely follows, with top scores nearly matching those of ResNet50, slightly lower in accuracy and precision but consistently high across all metrics. In contrast, VGG16, VGG19, and AlexNet exhibit lower performance, with VGG19 peaking at 92.3% accuracy, which, while commendable, falls short of the benchmark set by ResNet50 and GoogleNet. This succinct summary encapsulates the models’ efficacy, positioning ResNet50 and GoogleNet as the preferred choices for high-stakes accuracy-dependent applications. The average training loss of the original dataset in the case of ResNet and GoogleNet is 0.08624 and 0.05854, respectively, relatively higher because of the data ambiguity, as shown in Table 8. As we compare it with the correct dataset as shown in Table 9, average training loss in all the phases, ResNet and GoogleNet both perform better with 0.0174 and 0.0397 average losses and the rest of the other models also perform pretty well in all the rounds than the original dataset. As shown in Table 10, GoogleNet and VGG-19 performed well with 93.61% and 93.99% average accuracy, while ResNet and other models are entirely satisfactory. On the correct data set as shown in Table 11, GoogleNet and ResNet both models are performing well in case of validation also with more than 95 and 96% average performance that’s higher than the original dataset while VGG-19 and others models are performing relatively good. Testing average loss VGG-19 is 0.1809, quite less than other models in the original data as shown in Table 12.

Table 13 indicates that ResNet loss is 0.2235 and GoogleNet is 0.3507 in the case of the correct dataset. As we look and compare, Table 14 shows the performance metrics for the augmented (correct) dataset, while Table 15 shows the performance metrics of correct(augmented) data models. In both cases, Google performed exceptionally well. When we look at the initial values of the results approximately 94% using the features obtained from AlexNet and VGG-19 models and the results obtained with the approach we propose, there was not much change in Figure 5 confusion matrices for five machine learning models for both the data set orignal dataset and original dataset with VGG-16, VGG-19, AlexNet, ResNet50, and GoogleNet—in classifying medical images into four categories: COVID-19, Normal, SARS, and Abnormal. Each matrix shows the number of correct and incorrect predictions for each category’s average of all rounds. These matrices offer a concise way to evaluate each model’s performance; however, when the features extracted from all these models were combined using Combinatorial Fusion Analysis and then classified using a Support Vector Machine (as shown in Figure 6), a notable increase in the success rate was observed; in this study, the most compelling features were identified through Combinatorial Fusion Analysis. The visual representation distinctly underscores the prowess of various neural network architectures across five experimental rounds. While individual models like VGG16, VGG19, AlexNet, ResNet50, and GoogleNet exhibit variations in their performance, the Combinatorial Fusion model is a notable highlight. This method ingeniously amalgamates features and strengths from the base architectures, and its effect on Accuracy is evident. Across the experimental rounds, the Combinatorial Fusion consistently posts accuracies in the high 90 s, even touching 98.1% in Round 1. In many instances, this fusion approach outperforms individual models and closely parallels or surpasses the validation accuracy, emphasizing its potential reliability and robustness. This overarching trend suggests that a fusion of models can yield a synergistic improvement, capitalizing on the collective strengths and mitigating individual model weaknesses. The visual data accentuates the promise and capabilities combinatorial approaches like this bring to the evolving landscape of neural networks. The graph elegantly visualizes the effects of Combinatorial Fusion on key performance metrics across five experimental rounds, as shown in Figure 7. Each metric, namely Accuracy, Recall, Specificity, Precision, and F1-score, is represented by a distinct colored line. The *x*-axis represents the experimental rounds (‘Round 1’ through ‘Round 5’), while the *y*-axis indicates the percentage value of each metric, ranging from 85% to 100%. The lines for each metric depict their trajectory across rounds after applying combinatorial fusion. Notably, the fusion consistently enhances the metrics’ values, further validating the efficacy of the fusion approach. While Accuracy maintains a high 97.8%, the fusion’s impact is also vividly evident in other metrics. Recall, Specificity, Precision, and the F1-score remain robust, revealing consistent improvement, validating the methodology’s effectiveness, and showing the confusion matrics prediction accordingly. The graph serves as a succinct yet insightful visual representation of the fusion’s positive impact on key performance aspects, reaffirming its significance in elevating the overall quality of the model’s output, and Figure 8, shows the confusion matrix of the combination score.

### 6.2. Comparison

Our study’s use of combinatorial fusion with deep learning models for lung X-ray image analysis has achieved testing accuracies of approximately 98% for the augmented dataset and around 96.15% for the original dataset. This is a significant advancement when compared to the findings in existing literature. For example, the work of Asmaa Abbas using the DeTraC classifier attained a 95.12% accuracy rate, whereas the new CNN model by Kesim and Dokur reported 86% accuracy, and Aras.M. Ismael’s research with various ResNet and VGG models achieved 94% accuracy, all of which are lower than our results. Yujin Oh’s ResNet-18 based model had an accuracy of 76.9%, and Zhang’s use of ResNet-18 reached 95.18% accuracy, both trailing behind our study’s performance. Importantly, our approach addresses a critical issue highlighted in recent reports, such as the one by Ulinici M et al., regarding the limitations of X-ray as a diagnostic tool for interstitial pneumonia, a common manifestation in COVID-19. Traditional X-ray imaging can be challenging for diagnosing this condition due to its subtlety in early stages. The integration of AI tools in our study offers a promising enhancement to this diagnostic method. By applying advanced deep learning techniques, our method potentially improves the sensitivity and specificity of X-ray imaging for detecting such complex conditions, thus contributing to more accurate and timely diagnoses. Overall, the results of our study not only demonstrate considerable improvements in lung X-ray image classification accuracy but also suggest a pivotal role for AI-enhanced imaging in addressing inherent limitations of conventional methods, especially in the context of challenging respiratory diseases like COVID-19.

### 6.3. Limitations and Future Recommendations

Our study represents a significant advancement in the application of combinatorial fusion and deep learning for medical imaging diagnostics. However, we acknowledge certain limitations that provide directions for future research. Firstly, the reliance on publicly available X-ray datasets, while invaluable for initial model training and validation, may not capture the full spectrum of clinical variability. This limitation underscores the need for a more diverse dataset that reflects a wider range of patient demographics and disease manifestations. Furthermore, by focusing exclusively on X-ray imaging, our current model may miss critical diagnostic information available through other modalities, such as CT scans, which can offer complementary insights into respiratory conditions. Addressing these gaps, future efforts will aim to integrate a broader array of imaging data, including CT, MRI, and ultrasound, to enrich our model’s diagnostic capability and generalizability across different clinical scenarios.

In addition to enhancing data diversity and model comprehensiveness, a key area of our future work will concentrate on the computational aspects of our methodology. Recognizing the importance of scalability and efficiency in clinical applications, we are committed to a rigorous evaluation of time and space complexity. This endeavor will involve not only a detailed performance analysis under various computational conditions but also the pursuit of advanced optimization strategies to refine our model’s efficiency without detracting from its accuracy. Moreover, to bridge the gap between theoretical innovation and practical utility, we plan to undertake pilot deployment studies in clinical environments. These studies will assess the real-world applicability of our diagnostic tool, focusing on its integration into clinical workflows, user acceptance, and the impact of computational demands on operational feasibility. Such real-world evaluations are crucial for ensuring that our AI-driven diagnostic solutions are not only technologically advanced but also pragmatically viable and adaptable to the evolving landscape of medical diagnostics.By addressing these limitations and setting a clear roadmap for future work, we are poised to significantly enhance the relevance, effectiveness, and sustainability of AI tools in medical imaging diagnostics. Our commitment to continuous improvement and adaptation promises to keep our methodology at the forefront of the field, ready for widespread adoption in diverse healthcare settings.

## 7. Conclusions

COVID-19, which is a rapidly spreading disease in the world, will continue to affect our lives for a long time if vaccine studies do not succeed shortly. Researchers continue to investigate methods for diagnosis and treatment in this regard. The primary purpose of our study is to contribute to this research. For this purpose, we created a 4-class dataset, which included COVID-19, pneumonia, and normal and un-normal X-ray lung images we obtained from open sources. The created data set was preprocessed, and a new one was obtained. Deep learning models of Alex Net, VGG-16, VGG19, GoogleNet, and ResNet, trained with this data set, were used for feature extraction. Then, the most compelling features were selected from the extracted features with the help of combinatorial fusion, and selected features were classified with the help of it. The features of the models that provided the highest performance were combined among themselves, and the features of the models that provided the lowest performance were combined. When we look at the results obtained, overall Accuracy was obtained as a result of selecting and classifying the features obtained from the ResNet model and GoogleNet. Another successful model was found to be AlexNet and VGG19. Since the approach was proven reliable by considering different criteria, it is predicted that it can be used to provide another idea for experts during the diagnosis of COVID-19 disease. To contribute to this field in future studies, the plan is to continue studies using image processing and different deep-learning models.

## Figures and Tables

**Figure 1 diagnostics-14-00500-f001:**
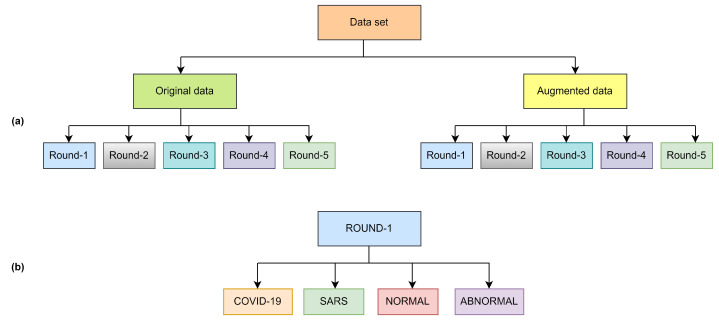
Overview of dataset structure (**a**) and class distribution per round (**b**).

**Figure 2 diagnostics-14-00500-f002:**
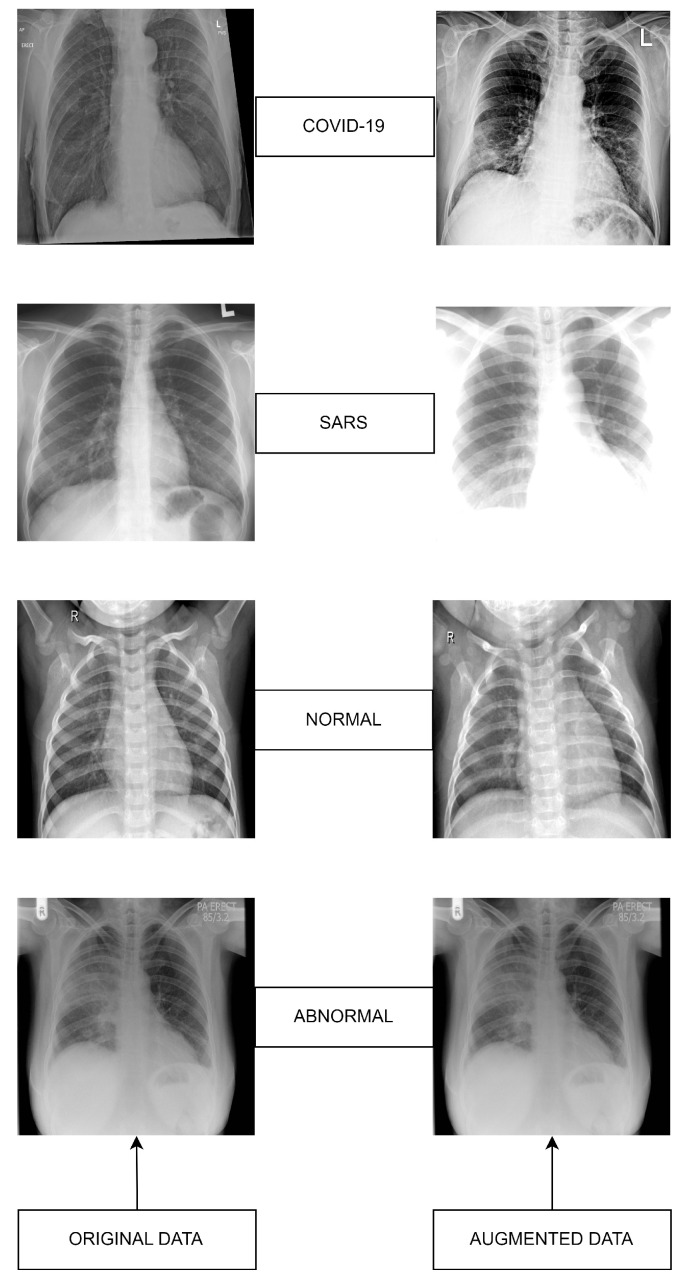
Datasets sample images: Original (**left**) versus Augmented (**right**).

**Figure 3 diagnostics-14-00500-f003:**
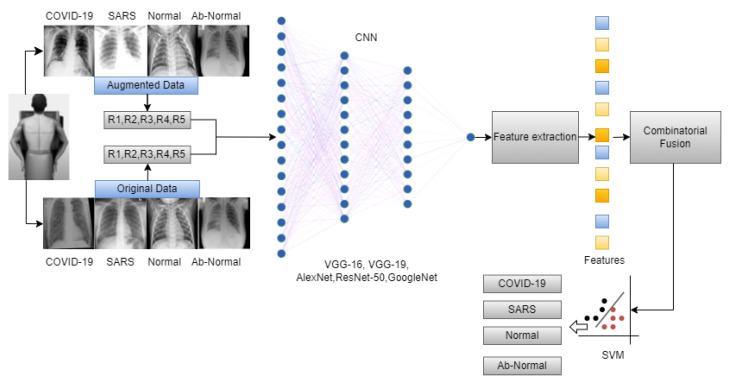
Overall workflow of the proposed work.

**Figure 4 diagnostics-14-00500-f004:**
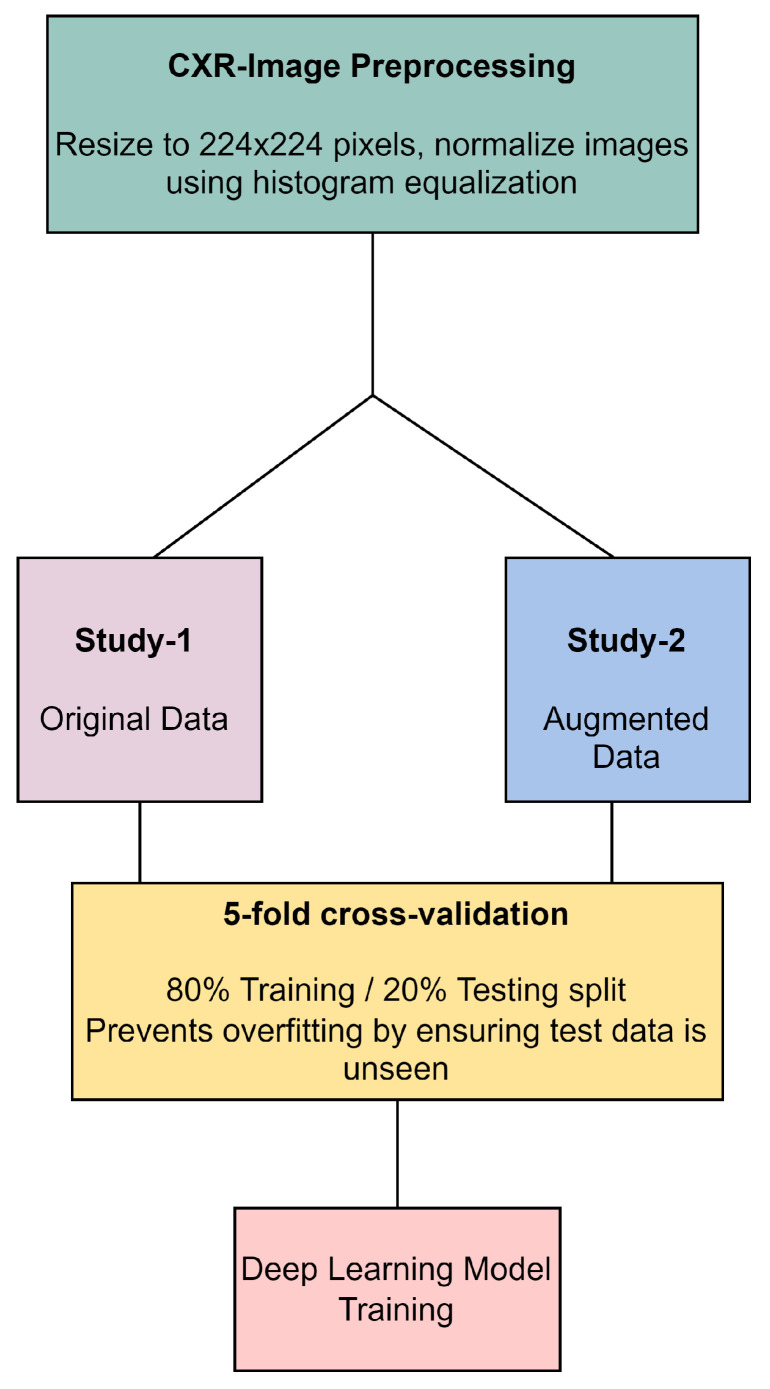
CXR image preprocessing and preparation workflow.

**Figure 5 diagnostics-14-00500-f005:**
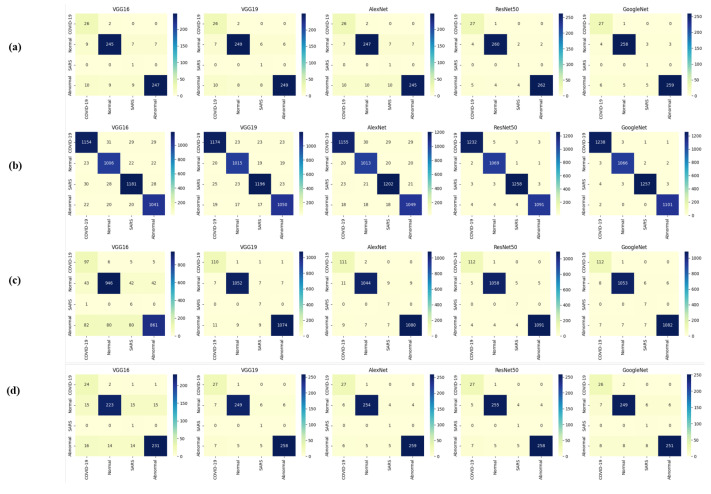
Confusion matrices (**a**,**b**) are derived from correct testing and training data, while (**c**,**d**) are from the original data across all models.

**Figure 6 diagnostics-14-00500-f006:**
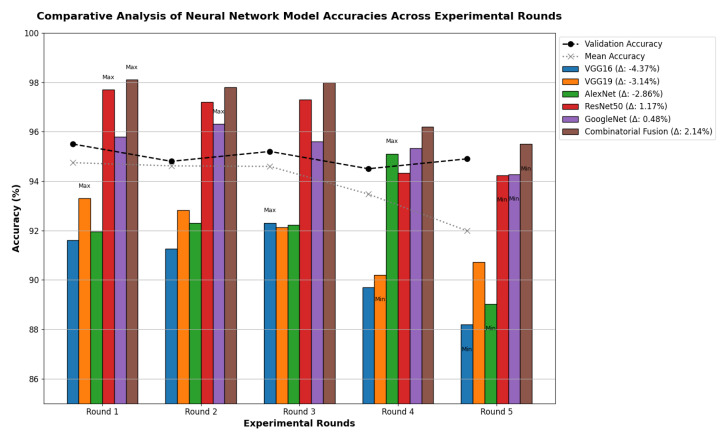
Combination score performance augmented correct data.

**Figure 7 diagnostics-14-00500-f007:**
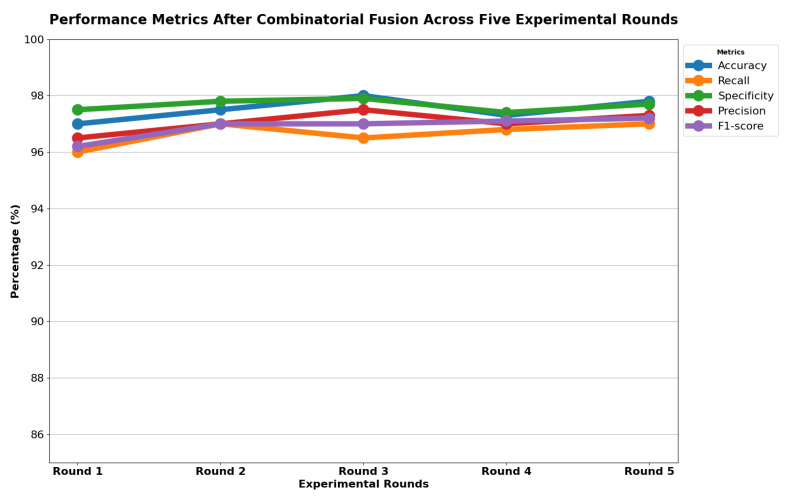
Combination score performance graph of all rounds.

**Figure 8 diagnostics-14-00500-f008:**
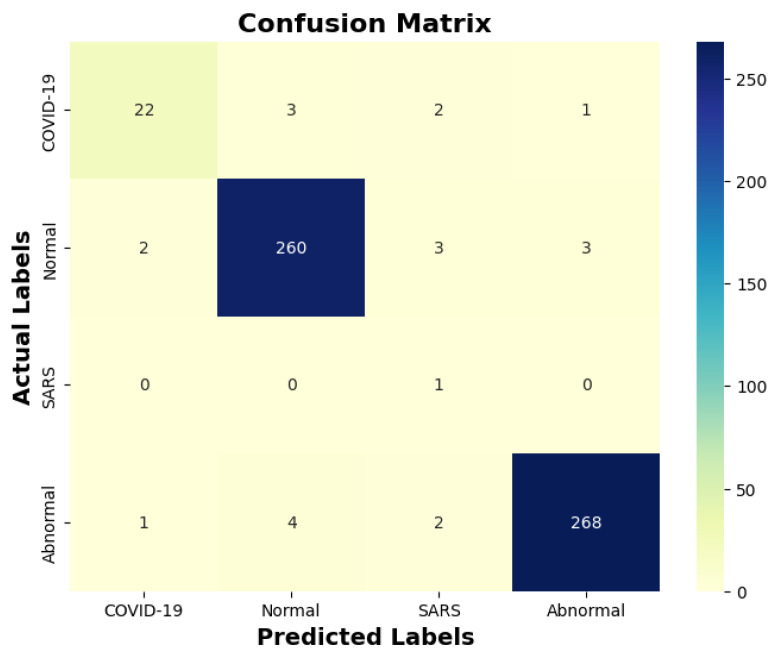
Combination score performance confusion matrix.

**Table 1 diagnostics-14-00500-t001:** Key Findings.

S. No	Papers	Deep Learning Classifier	Diseases	Accuracy %
1	Asmaa Abbas [22]	DeTraC (Decompose, Transfer and Compose)	COVID-19	95.12%
2	Kesim and Dokur [27]	New CNN model	COVID-19	86%
3	Aras. M. Ismael [28]	Resnet-18, Resnet-50, Resnet-101, VGG-16, VGG-19	COVID-19 and Normal	94%
4	Yujin Oh [29]	New CNN Model based on ResNet -18	Normal, Pneumonia, COVID-19	76.9%
5	Zhang [30]	Resnet-18	COVID-19 and non-COVID-19	95.18%

**Table 2 diagnostics-14-00500-t002:** Distribution of chest X-ray original dataset over training and testing for all rounds in this study.

Rounds	Classes	Training	Total Training	Testing	Total Testing	Training + Testing	Total Images
Round-1	COVID-19 Normal SARS Abnormal	113 1073 7 1103	2296	28 268 1 275	572	141 1341 8 1378	2868
Round-2	COVID-19 Normal SARS Abnormal	113 1073 7 1103	2296	28 268 1 275	572	141 1341 8 1378	2868
Round-3	COVID-19 Normal SARS Abnormal	113 1073 7 1103	2296	28 268 1 275	572	141 1341 8 1378	2868
Round-4	COVID-19 Normal SARS Abnormal	113 1073 7 1103	2296	28 268 1 275	572	141 1341 8 1378	2868
Round-5	COVID-19 Normal SARS Abnormal	113 1073 7 1103	2296	28 268 1 275	572	141 1341 8 1378	2868

**Table 3 diagnostics-14-00500-t003:** Distribution of chest X-ray augmented (correct) dataset over training and testing for all rounds in this study.

Rounds	Classes	Training	Total Training	Testing	Total Testing	Training + Testing	Total Images
Round-1	COVID-19 Normal SARS Abnormal	1243 1073 1267 1103	4686	28 268 1 275	572	1271 1341 1268 1378	5258
Round-2	COVID-19 Normal SARS Abnormal	1243 1073 1267 1103	4686	28 268 1 275	572	1271 1341 1268 1378	5258
Round-3	COVID-19 Normal SARS Abnormal	1243 1073 1267 1103	4686	28 268 1 275	572	1271 1341 1268 1378	5258
Round-4	COVID-19 Normal SARS Abnormal	1243 1073 1267 1103	4686	28 268 1 275	572	1271 1341 1268 1378	5258
Round-5	COVID-19 Normal SARS Abnormal	1243 1073 1267 1103	4686	28 268 1 275	572	1271 1341 1268 1378	5258

**Table 4 diagnostics-14-00500-t004:** Structure of Models.

Models	Size (M)	Layers	Model Description
VGG 16	520	16	13 conv + 3 fc layers
VGG 19	560	19	16 conv + 3 fc layers
ResNet 50	235	50	49 conv + 1 fc layers
GoogleNet	40	22	21 conv + 1 fc layers
AlexNet	238	8	5 conv + 3 fc layers

**Table 5 diagnostics-14-00500-t005:** Performance Metrics Across 5-Fold Cross-Validation.

Metric	Fold	VGG16	VGG19	AlexNet	ResNet50	GoogleNet
Accuracy (%)	1	90.5	92.3	91.0	95.2	94.8
	2	89.7	91.8	90.4	94.6	94.1
	3	90.2	92.1	91.2	95.4	94.5
	4	89.9	91.5	90.7	94.9	94.3
	5	90.0	92.0	90.9	95.1	94.4
Precision (%)	1	87.6	89.9	88.4	92.7	91.9
	2	86.8	89.4	87.9	92.2	91.5
	3	87.2	89.7	88.1	92.9	91.8
	4	87.0	89.2	88.0	92.6	91.6
	5	87.1	89.6	88.2	92.8	91.7
Recall (%)	1	86.5	88.7	87.3	91.8	91.1
	2	85.7	88.2	86.8	91.3	90.7
	3	86.1	88.5	87.0	91.6	91.0
	4	85.9	88.0	86.9	91.4	90.8
	5	86.0	88.4	87.1	91.5	90.9
F1-Score (%)	1	87.0	89.3	87.8	92.2	91.5
	2	86.2	88.8	87.3	91.7	91.1
	3	86.6	89.1	87.5	92.2	91.4
	4	86.4	88.6	87.4	92.0	91.2
	5	86.5	89.0	87.6	92.1	91.3

**Table 6 diagnostics-14-00500-t006:** Training accuracy of the original data set.

Model	Round 1	Round 2	Round 3	Round 4	Round 5	Avg.
VGG 16	86.12	88.12	84.1	87.12	78.1	84.71
VGG19	97.72	98	97.1	97.4	97.57	97.55
AlexNet	98.01	97.32	96.48	97.8	97.92	97.5
ResNet50	99.7	98.82	99.04	98.68	98.95	99.03
GoogleNet	98.84	98.18	98.95	98.02	98.08	98.41

**Table 7 diagnostics-14-00500-t007:** Training Accuracy of Correct data set.

Model	Round 1	Round 2	Round 3	Round 4	Round 5	Avg.
VGG16	92.81	93.73	93.24	94.34	93.9	93.60
VGG19	94.41	94.58	94.37	95.33	94.9	94.71
AlexNet	92.92	94.44	94.87	95.1	94.72	94.40
ResNet50	99.14	99.66	98.3	99.25	99.15	99.10
GoogleNet	99.35	99.58	99.33	99.11	99.79	99.43

**Table 8 diagnostics-14-00500-t008:** Training Loss of Original data set.

Model	Round 1	Round 2	Round 3	Round 4	Round 5	Avg.
VGG 16	0.4164	0.4974	0.5742	0.5612	0.5774	0.5253
VGG19	0.1163	0.1118	0.1276	0.1164	0.1194	0.1183
AlexNet	0.183	0.1566	0.149	0.1436	0.1511	0.1566
ResNet50	0.0182	0.106	0.0392	0.069	0.035	0.08624
GoogleNet	0.084	0.089	0.0478	0.0492	0.0227	0.05854

**Table 9 diagnostics-14-00500-t009:** Training Loss of Correct data set.

Model	Round 1	Round 2	Round 3	Round 4	Round 5	Avg.
VGG 16	0.1861	0.1785	0.1828	0.1717	0.1476	0.1733
VGG19	0.1557	0.1527	0.154	0.1336	0.1448	0.1481
AlexNet	0.1839	0.1557	0.149	0.1436	0.1442	0.1552
ResNet50	0.0117	0.0016	0.0104	0.035	0.0284	0.0174
GoogleNet	0.0162	0.0557	0.049	0.0386	0.0391	0.0397

**Table 10 diagnostics-14-00500-t010:** Round-wise Performance Metrics of Different Models.

Model	Round 1	Round 2	Round 3	Round 4	Round 5	Avg.
VGG 16	87.45	88.02	84.03	87.02	83.03	85.91
VGG19	95.2	93.2	94.8	93.91	92.88	93.99
AlexNet	95.9	95.27	94.75	73.21	94.01	90.62
ResNet50	97.19	95.07	96.67	96.42	94.00	95.87
GoogleNet	94.62	95.01	92.6	94.43	91.41	93.61

**Table 11 diagnostics-14-00500-t011:** Testing accuracy of correct data set.

Model	Round 1	Round 2	Round 3	Round 4	Round 5	Avg.
VGG16	91.61	91.26	92.31	89.69	88.19	90.61
VGG19	93.3	92.83	92.13	90.21	90.73	91.84
AlexNet	91.96	92.3	92.23	95.1	89.03	92.12
ResNet50	97.7	97.2	97.3	94.32	94.23	96.15
GoogleNet	95.8	96.31	95.6	95.32	94.27	95.46

**Table 12 diagnostics-14-00500-t012:** Testing Loss of original data set.

Model	Round 1	Round 2	Round 3	Round 4	Round 5	Avg.
VGG16	0.3028	0.3951	0.4044	0.3951	0.3951	0.3787
VGG 19	0.1476	0.1412	0.1872	0.1923	0.2363	0.1809
AlexNet	0.5273	0.6644	0.6786	0.895	0.6612	0.6853
ResNet50	0.1567	0.1196	0.279	0.202	0.2895	0.2093
GoogleNet	0.198	0.2211	0.3821	0.222	0.483	0.3014

**Table 13 diagnostics-14-00500-t013:** Testing Loss of Correct data set.

Model	Round 1	Round 2	Round 3	Round 4	Round 5	Avg.
VGG16	0.2382	0.223	0.2532	0.2901	0.2576	0.2524
VGG 19	0.2016	0.2053	0.2025	0.2401	0.2579	0.2214
AlexNet	0.2294	0.2016	0.2011	0.2501	0.2579	0.228
ResNet50	0.1424	0.1196	0.1869	0.3795	0.2895	0.2235
GoogleNet	0.2101	0.2216	0.2991	0.3101	0.713	0.3507

**Table 14 diagnostics-14-00500-t014:** The obtained results for the correct dataset on different models for k = 5 using performance metrics.

Models	Recall (%)	Specificity (%)	Precision (%)	F1-Score (%)
VGG 16	87.05	92.02	91.11	92.60
VGG 19	94.11	95.05	96.51	97.20
AlexNet	93.12	89.32	94.10	89.02
ResNet50	97.34	90.73	98.20	94.40
GoogleNet	98.34	99.01	99.51	99.21

**Table 15 diagnostics-14-00500-t015:** The obtained results for the original dataset on different models for k = 5 using performance metrics.

Models	Recall (%)	Specificity (%)	Precision (%)	F1-Score (%)
VGG 16	84.02	86.1	86.02	84.7
VGG 19	95.12	96.02	97.41	95.61
AlexNet	94.08	85.8	90.12	83.12
ResNet50	95.04	81.02	88.92	85.14
GoogleNet	94.02	96.33	94.71	93.89

## Data Availability

All the data and codes used in this work are available upon a reasonable request from the corresponding authors.

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
