# Peer review of "Improving Respiratory Infection Diagnosis with Deep Learning and Combinatorial Fusion: A Two-Stage Approach Using Chest X-ray Imaging"

_diagnostics, 2024, doi:10.3390/diagnostics14050500_

Round 1
Reviewer 1 Report
Comments and Suggestions for Authors
The article needs massive revision to meet the quality level required by the esteemed journal. The following points need to be addressed in the revision:
- The title needs improvement as it does not represent the objective described in the abstract. The word “analytics” also needs to be checked again.
- All the results need to be checked in their code. I felt like augmentation was not properly applied to the dataset.
- Two stage data analytic approach is not clear along with the combinatorial fusion.
- “We address data imbalance through five-fold validation by utilize…” is against the norms of machine learning principles.
- The main innovation and contribution of this research should be clarified in the abstract and introduction.
- Please thoroughly check your article for typos and grammatical mistakes.
- Tables have not been properly explained and discussed in the text.
- Figure number is absent in Line number 263.
- I could not find the figure representing the methodology of your work showing different architectures being used. Figure 1 is not able to solve this issue.
- SVM (Line 300) has been used as the only classifier in this study. You need to add more classifiers. Why you are not using the deep learning algorithms in your study as you mentioned in the abstract?
- Citation of the work of others is missing in the article, e.g. line numbers 305-310.
- I am not satisfied with the analysis presented in this work. The results should be related to 5-fold type validation.
- Time and space complexity is absent.
- Rewrite the conclusions in the light of the modifications/improvements suggested.
- Comparison with SOTA techniques is absent.
- Add a subsection related to the “limitations and future recommendations” before the Conclusions.
Comments on the Quality of English Language
Thoroughly check your article for typos and grammatical mistakes.
Author Response
We have addressed all your comments and made the necessary changes. Please see the attachment. Thank you for your valuable feedback.

Reviewer 2 Report
Comments and Suggestions for Authors
Thank you for the possibility to review the manuscript titled: “Enhancing Early Diagnosis of Respiratory Infections through Two-Stage Analytics with Deep Neural Networks and Combinatorial Fusion on Chest X-ray Imaging”. The manuscript is interesting and easy to read. The studies aim is to analyze X-ray images through a special algorithm enhances with AI. The aim and methods of the study are understandable, however there are multiple structural mistakes in the manuscript. Therefore, I would recommend:
-Please review the language of the manuscript. There are multiple type mistakes. For instance:
“and Pneumonia” (pneumonia should be written with a lower case); “the presence of [4].” (the end of the sentence is missing); “The overall workflow of this study is provided in Figure ??.” (what figure?). Therefore, please carefully revise the text.
-Results and discussions should be written separately. Don non analyses the results before discussion section.
- Recheck all of the references. For instance:
9. et al., F.M.S. A comprehensive survey of COVID-19 detection using medical images. SN Computer Science 2021, 2, 1–22. 413
10. et al., E.S. Augmented bladder tumor detection using deep learning. European urology 2019, 76, 714–718.
Reference 9 and 10 contain mistakes.
-Compare the results with other manuscripts. For instance there have been multiple reports that X-ray is a weak diagnostic tool for interstitial pneumonia (Ulinici M et al. Screening, Diagnostic and Prognostic Tests for COVID-19: A Comprehensive Review. Life (Basel). 2021 Jun 14;11(6):561. doi: 10.3390/life11060561. PMID: 34198591; PMCID: PMC8231764). Enhancement of this diagnostic method with AI tools might be helpful.
Please take into account the recommendation in the spirit of improving the quality of submission.
Comments on the Quality of English LanguageThere are multiple type mistakes in the text
Author Response

(The authors gave the same response as above.)

Reviewer 3 Report
Comments and Suggestions for Authors
Dear authors, I find your idea very interesting and relevant. However, I need to seek clarification and would like to offer some suggestions as follows. Thank you.
-In my humble opinion, the background can be shortened, especially in the initial part where a synthesis of COVID-19 and diagnostic capabilities is provided. A lengthy text may risk losing attention. Additionally, RT-PCR currently remains the gold standard for Sars-CoV-2 diagnosis (Abdullah A, Sam IC, Ong YJ, Theo CH, Pukhari MH, Chan YF. Comparative Evaluation of a Standard M10 Assay with Xpert Xpress for the Rapid Molecular Diagnosis of SARS-CoV-2, Influenza A/B Virus, and Respiratory Syncytial Virus. Diagnostics (Basel). 2023 Nov 22;13(23):3507. doi: 10.3390/diagnostics13233507. PMID: 38066748; PMCID: PMC10706428.). However, your analysis focuses on the diagnosis of Sars-CoV-2 pneumonia or COVID-19 pneumonia, which can be conducted through imaging but typically in a later phase of the infection.
-Please complete the sentence at line 33. The sentence that begins at line 37 also appears incomplete. Could you please finish it?
-In paragraph 2, I would kindly request, if possible, to list (1,2,3, or a,b,c) all the works you have considered, facilitating a more straightforward reading. Thank you.
-At line 137, when you refer to SARS, do you mean non-COVID pneumonia or ARDS? Please clarify this aspect. Thank you.
-In the text, you mention the four classes, namely COVID, SARS, normal, and non-normal. Could you clarify whether, by SARS, you are referring to non-COVID pneumonia or ARDS?
-In my humble opinion, the sentence at line 167 should be reviewed or excluded.
-In Table 3, in rows 'round 4' and 'round 5' under the columns 'training' and 'testing,' what do the four vertically aligned numbers represent? Please clarify these data.
-At line 199, it is not specified which figure is being referenced.
-At line 263, it is not specified which figure is being referenced.
-At line 277, the X-rays for SARS are stated as 1276, while in Table 3, they appear as 1267. Could you please correct this inconsistency?
-Could you please improve the readability of Figure 6 left?
-In my humble opinion, adding captions to the Tables could enhance comprehension.
-I consider the analysis and discussion to be valid. The conclusion is comprehensive.
-The bibliography is very comprehensive and relevant. Please, could you correct the bibliographic entries 5, 9, 10, 16, 21, 33. Thank you.
Comments on the Quality of English LanguagePlease complete the sentence at line 33. The sentence that begins at line 37 also appears incomplete. Could you please finish it?
Author Response

(The authors gave the same response as above.)

Round 2
Reviewer 1 Report
Comments and Suggestions for Authors
The paper is in good shape now. The following points still need to be addressed as a minor revision:
- Tables 2, 3, and 4 have repeated values. You should avoid this massive repetition. Table 4 has the same values for the deep learning architectures. Why not give it a text form?
- The augmented data is avoided as the test data. When I go through your article, I have the feeling from Fig 1 and the text around that this violation has occurred. Please double-check it and correct it. Fig 1 does not represent the cross-validation use.
- The classifier seems absent in the article that was used to classify the data depending on features of different types.
- The tables of combinatorial fusion are absent. The SVM parameterization and its type are missing.
- The training results are not worth mentioning in the abstract. Remove the training results from the abstract.
- Mention the two stages idea in Figure 1 and its associated text.
Minor modifications at the proof stage will do the job.
Author Response
Thank you for your valuable comment, We have addressed the corrections as per your valuable suggestions. please find the attachment-

Reviewer 2 Report
Comments and Suggestions for Authors
The authors have met all of the necessary corrections. There is only one missing reference which is cited in the text but not in the reference section (Ulinici M et al. Screening, Diagnostic and Prognostic Tests for COVID-19: A Comprehensive Review. Life (Basel). 2021 Jun 14;11(6):561. doi: 10.3390/life11060561. PMID: 34198591; PMCID: PMC8231764.)
Author Response

(The authors gave the same response as above.)

Reviewer 3 Report
Comments and Suggestions for Authors
Thank you for responding to my considerations. Congratulations on the publication.
Author Response
Thank you for your insightful feedback and constructive suggestions. In response, we have thoroughly revised our manuscript to incorporate these recommendations, ensuring a more comprehensive and polished presentation.